# Microbial and Physico-Chemical Characterization of Cold Smoked Sea Bass (*Dicentrarchus labrax*), a New Product of Fishery

**DOI:** 10.3390/foods12142685

**Published:** 2023-07-12

**Authors:** Michela Pellegrini, Lucilla Iacumin, Jelka Pleadin, Greta Krešić, Elisabetta Orecchia, Andrea Colautti, Ana Vulić, Nina Kudumija, Cristian Bernardi, Giuseppe Comi

**Affiliations:** 1Department of Agricultural, Food, Environmental and Animal Science, University of Udine, 33100 Udine, Italy; pellegrini.michela@spes.uniud.it (M.P.); lucilla.iacumin@uniud.it (L.I.); orecchia.elisabetta@spes.uniud.it (E.O.); colautti.andrea@spes.uniud.it (A.C.); 2Laboratory for Analytical Chemistry, Croatian Veterinary Institute, Savska Cesta 143, 10000 Zagreb, Croatia; pleadin@veinst.hr (J.P.); vulic@veinst.hr (A.V.); kudumija@veinst.hr (N.K.); 3Department of Food and Nutrition, Faculty of Tourism and Hospitality Management, University of Rijeka, Primorska 46, 51410 Opatija, Croatia; gretak@fthm.hr; 4Department of Veterinary Medicine and Animal Sciences, University of Milan, 20122 Lodi, Italy; cristian.bernardi@unimi.it

**Keywords:** cold smoked sea bass, physico-chemical characteristics, microbial characteristics

## Abstract

The aim of this study was to investigate the microbial and physico-chemical characteristics of cold smoked sea bass (CSSB), a novel italian fish product. The microbiological analyses showed the presence of bacterial contamination from the raw material, the environment, and the production process. The microbial spoilage population was dominated by lactic acid bacteria (LAB) associated with Gram-negative fermenting bacteria, including *Photobacterium phosphoreum* and psychrotrophic Enterobacteriaceae. *Brochotrix thermospacta* and *Aeromonas* spp. were also present; in contrast, mould and yeast were not detected (<2 CFU/g). High levels (6–7 log CFU/g) of LAB and total bacteria count (TBC) were observed from day 45 of storage; however, their presence does not seem to have influenced the total volatile basic nitrogen (TVB-N), which always remained below 35 mg N/100 g. Consequently, the product is acceptable until day 60 of storage, considering that the malonaldehyde index (TBARS) was lower than 6.5 nmol/g. Pathogenic bacteria such as *Salmonella* spp. and *Listeria monocytogenes* were not detected. Currently, there is a growing demand for seafood due to its high quality and nutritional value. Cold smoked sea bass offers a source of macro- and micronutrients essential for the proper functioning of the human body. It is also rich in protein and omega-3 fatty acids. The WHO and FAO evaluated the benefits and risks and concluded that there is convincing evidence of health benefits from fish consumption, such as a reduction in the risk of heart failure and improved neurodevelopment in infants and young children when fish is consumed by the mother before and during pregnancy. The CSSB analysed in this study demonstrated to have health benefits due to long-chain omega-3 PUFAs and other nutrients, such as proteins, minerals, and vitamin D, which are sometimes difficult to obtain from other sources. The results show that CSSB has a high nutritional value and excellent microbial quality.

## 1. Introduction

Fish is an excellent source of nutrients and has a high nutritional value for humans [1]. However, due to its high water activity (Aw), neutral pH, relatively high contents of nitrogen compounds and free amino acids, and presence of autolytic enzymes, it is very susceptible to microbiological and chemical spoilage [2]. Different methods have been developed and are currently used to extend the shelf life of fish and fishery products; however, for fresh fish, rapid cooling and storage on ice increases shelf life by only 9–12 days, despite the use of innovative packaging and technological methods [3,4]. For fish products, the most commonly used preservation method is smoking, a traditional technology used for centuries to extend the shelf life of fresh fish [4]. The main purpose of smoking is to confer an appealing appearance and taste and to decrease microbial growth.

Cold smoked fish (CSF) are largely produced all over the world. Salmon and trout are the main fish subjected to salting and cold smoking to increase their organoleptic characteristics and extend their shelf life. Recently, due to consumer demand, other fish, such as sea bass, sardines and dolphinfish, have also been treated with the aforementioned techniques [5,6]. The use of cold smoking can provide new products with high nutritional and added value [6]. In Italy, to meet market demand, a company experienced in smoking fish products has started to produce cold smoked sea bass (CSSB) fillets.

Sea bass (*Dicentrarchus labrax*) is a typical aquaculture product that is extensively bred in Mediterranean areas, and Greece, Turkey, Italy, Spain, Croatia, and Egypt are the largest producers [7,8].

Fish flesh is considered a functional food because of its high nutritional value, as it is rich in essential proteins and lipids. In particular, the lipid fraction includes polyunsaturated fatty acids (PUFAs), such as docosahexaenoic acid (DHA), eicosapentaenoic acid (EPA), and arachidonic acid (ARA) [9,10]. Among them, n-3 long-chain polyunsaturated fatty acids (n-3 LCPUFAs), due to their anti-inflammatory and cardioprotective effects [8], are particularly appreciated by consumers. Fish flesh can also contribute to the intake of some essential nutrients, such as iodine, calcium, selenium, and vitamin D. Regular fish consumption could prevent several diseases such as obesity, micronutrient deficiencies, cardiovascular diseases, heart failure, and stroke [10], and according to Arab-Tehrany et al. [11], n-3 PUFA intake is associated with a lower risk of depression.

The cold smoking process includes salting, dehydration, and smoking. Salting and dehydration modify the water activity (Aw) and inhibit bacteria and mould [12,13]. In addition, smoke is rich in phenols and, therefore, has a bacteriostatic effect. However, the temperature used during the smoking process is less than 30 °C, and for this reason, there is no effect on the microbial population. Consequently, the CSF may be subjected to microbial spoilage during storage at 4 °C and are classified under lightly preserved fish products (LPFPs) due to their specific chemical properties, represented by a water content of approximately 74%, a NaCl concentration in the water phase ranging from 3.5 and 5.5% (*w*/*w*), and a smoke treatment corresponding to 0.6 mg of phenol per 100 g of product [14]. Additionally, the pH value (from 6 to 6.3), vacuum packaging (VP), and storage temperature (lower than 5 °C) are not sufficient to prevent the spoilage or growth of pathogenic microorganisms [15,16]. Certain microorganisms can grow during storage, leading to the production of off-odours, off-flavours, and a pasty texture [16,17].

The microbial population of CSF is complex and heterogeneous, and it has been widely studied. VP or modified atmosphere packaging (MAP), which limits the available oxygen, promotes the predominance of anaerobic microorganisms such as lactic acid bacteria (LAB) [18,19], Gram-negative fermenting bacteria, mainly represented by *Photobacterium phosphoreum* and psychrotrophic Enterobacteriaceae [12,20], and other bacteria, including *Brochotrix thermospacta* and *Aeromonas* spp. Mould and yeast were rarely isolated, and the yeasts found throughout storage are always considered subdominant microbiota [17,20,21,22]. Cold smoked fish can also support the growth of pathogenic bacteria, and even, if rarely, this type of product has been associated with outbreaks due to the presence of *Salmonella*, *Staphylococcus aureus*, *Listeria monocytogenes*, and *Clostridium botulinum* [23,24]. To overcome these problems, in recent years, it was suggested to use bioprotective cultures to inhibit spoilage microorganisms and *L. monocytogenes* and to control *C. botulinum* through the water phase salt (WPS) concentration [12,16,25,26].

Based on the day of sampling, and in particular at the end of the shelf life, the contamination level of CSF can reach 8–9 log CFU/g, and it is always composed of LAB. This value is often reached 2 weeks before the end of the shelf life, but the product can still be acceptable because of the lack of sensory defects. Consequently, the microbial concentration cannot be used as a spoilage indicator for CSF [15].

It is well known that the initial microbiota of this type of product is dominated by Gram-negative bacteria such as *Photobacterium*, *Shewanella*, *Vibrio*, and *Yersinia*, which are typically associated with fresh raw fish. Conversely, during storage, Gram-positive bacteria, especially LAB, become the representative microbiota [17,20]. However, some strains of Enterobacteriaceae (*Serratia proteamaculans* and *Hafnia alvei*), *Photobacterium*, and *Brochothrix* can also be present at sufficient levels to induce spoilage [17,20].

Different strains of LAB were identified by culture-dependent and culture-independent techniques [17,27], providing a complete and clear overview of the microbial ecosystem composition and dynamics in CSF. The literature has demonstrated that LAB presence is often correlated with the product, the type of processing, or the production area [27].

The use of culture-independent methods confirmed that the initial microbiota was generally dominated by *Photobacterium phosphoreum/kishitanii, Aliivibrio* spp., *Lactobacillus sakei, B. thermosphacta* and, to a lesser extent, LAB, such as *Leuconostoc gasicomitatum*, *Lactococcus piscium*, *Carnobacterium divergens*, and *Carnobacterium maltaromaticum* [28]. The storage conditions exerted strong selective pressure on the initial microbiota, and at the spoiling date the composition can be highly variable [27].

The influence of the type of processing used is demonstrated by the presence and dominance of marine *Vibrio*, including *P. phosphoreum*, in incipient spoilage of normal dry-salted salmon, while a mixture of LAB and Enterobacteriaceae dominated the injection-brined salmon [29].

Wiernasz et al. [27] showed that some CSF were dominated by *B. thermosphacta* and LAB (*C. maltaromaticum*, *Lactobacillus fuchuensis*) or by *P. phosphoreum*/*kishitanii* alone. *Psychrobacter*, *Shewanella*, *Salinivibrio*, and, unexpectedly, *Pantoea* genera were also identified in CSF [27].

The different processing plants and smokehouses can also influence the microbial population of spoiled CSF [18].

It is well recognized that many of the microorganisms in CSF do not contribute to spoilage [15]. Leroi et al. [20] observed that despite frequent isolation, *P. phosphoreum* and most LAB are usually not involved in spoilage. Aerobic microorganisms such as *Shewanella putrefaciens* and *Pseudomonas* can also be isolated in CSF, but their growth is inhibited by VP [20,30]. Rarely, *S. putrefaciens* survived the smoking process and grew under anaerobic conditions [31]. According to various authors [19], only LAB should be considered a potential spoilage bacteria in vacuum-packed CSF.

The CSF production process is widely used in Italy and has a long history. Italian farmed fish can rely on good nutritional and safety standards [6,10] and must satisfy the general need to increase production with innovative products such as CSF.

Recently, among CSF, a new value-added product was proposed and developed by using sea bass (*D. labrax*). Considering that little is known about this CSF product, the aim of the study was to investigate the physico-chemical, microbial, and nutritional characteristics of CSSB fillets produced in the Friuli Venezia Giulia region (north-eastern Italy).

## 2. Materials and Methods

### 2.1. Sample Processing

The samples consisted of 3 different lots of gutted sea bass (*Dicentrarchus labrax*) weighing between 474 and 578 g and approximately 35 cm long. The fish were bred in sea cages by Orada Adriatic d.o.o. in Cres (Split, Croatia), collected and slaughtered after being bathed in water and ice. The sea bass were processed with a technique similar to that used for trout and salmon. Briefly, the fresh fish were filleted (baffe) and salted until they reached a WPS value greater than or equal to 3.5%; then, they were desalted and smoked at a temperature lower than 30 °C. After smoking, the fillets were vacuum packed in plastic bags (PE/PA, Niederwieser group, Italy), stored at 4 °C, and transferred to the Department of Agricultural, Food, Environmental and Animal Sciences of the University of Udine (Di4a). Each sample weighed approximately 200 g. Cold smoked sea bass fillets belong to the category of minimally processed foods, which are treated with preservation techniques that do not alter their original nutrient composition and overall quality [32].

### 2.2. Microbial Analysis

Each lot included 15 samples that were analysed at 0, 15, 30, 45, and 60 days (the typical deadline of the shelf life of CFS). To perform the storage test, the samples were stored at 4 ± 2 °C for 20 days and then at 8 °C till 60 days, which is the standard temperature of a supermarket refrigerator, until the end of storage (60 days). At each sampling time, three samples were subjected to microbiological analyses, which included the quantification of the total bacterial count (TBC) on gelatine sugar-free agar (Oxoid, Milan, Italy) incubated at 30 °C for 48–72 h; LAB on De Man, Rogosa, and Sharpe agar (MRS; Oxoid, Milan, Italy) incubated at 37 °C for 48 h (double layer method); yeasts and moulds on malt extract agar (MEA; Oxoid, Milan, Italy) incubated at 25 °C for 72–96 h; Enterobacteriaceae on violet red bile glucose agar (VRBGA; Oxoid, Milan, Italy) incubated at 37 °C for 24 h; coagulase-positive staphylococci on Baird–Parker agar (BP; Oxoid, Milan, Italy) supplemented with egg yolk tellurite emulsion (Oxoid, Milan, Italy) incubated at 35 °C for 24–48 h and confirmed by the coagulase test; sulfite-reducing clostridia on differential reinforced clostridial medium (DRCM; Merck, Darmstadt, Germany) incubated at 37 °C for 24–48 h in a jar prepared for anaerobic reaction with a gas-packing anaerobic system (BBL, Becton Dickinson, Franklin Lakes, NJ, USA). *L. monocytogenes* was detected and quantified according to the ISO method [33], and *Salmonella* spp. was detected and quantified according to the ISO method [34].

### 2.3. Microbial Identification

On each day of sampling, from the gelatine sugar-free agar, MRS, and VRBGA plates 250 colonies were selected independently of their morphology, colour, and size and isolated and purified on the same agar plates of growth. After purification, the colonies were identified according to the PCR-DGGE (PCR-denaturing gradient gel electrophoresis) method, as described by Iacumin et al. [35]. Briefly, amplicons to be subjected to DGGE analysis were obtained using primers P1 (5′-GCGGCGTGCCTAATACATGC-3′) and P4 (5′-ATCTACGCATTTCACCGCTAC-3′) spanning the V3 region of the 16S rDNA [23]. A GC-clamp (5′-CGCCCGCCGCGCCCCGCGCCCGTCCCGCCGCCCCCGCCCG-3′) was attached to the 5′ end of primer P1. Amplifications were carried out in a final volume of 25 μL containing 1 μL (100 ng total) template DNA, 10 mM Tris HCl (pH 8.3), 50 mM KCl, 1.5 mM MgCl_2_, 0.2 mM deoxynucleoside triphosphates (dNTPs), 1.25 U Taq polymerase (Invitrogen, Milan, Italy), and 0.2 μM of each primer using the C1000 touch thermal cycler (Bio-Rad, Milan, Italy). The amplification cycle included an initial denaturation step at 95 °C for 5 min, followed by 35 series composed of denaturation performed at 95 °C for 1 min, annealing at 45 °C for 1 min, and extension performed at 72 °C for 1 min. Finally, an extension cycle performed at 72 °C for 7 min was added. Electrophoresis was performed in a 0.8 mm thick polyacrylamide gel (8% (wt/vol) acrylamide bisacrylamide (37.5:1)), with a denaturing gradient from 30% to 50% (100% corresponded to 7 M urea and 40% (wt/vol) formamide) increasing in the direction of the electrophoretic run using the Dcode universal mutation detection system (Bio-Rad, Hercules, CA, USA). The gels were subjected to a constant voltage of 130 V for 3 h and 30 min at 60 °C. After electrophoresis, the gels were stained for 30 min in 1.25× tris-acetate-EDTA containing 1× SYBR Green (final concentration; Molecular Probes, Milan, Italy). Pictures of the gels were visualized under UV light using the Syngene G: Box Chemi-XX9 (Syngene, Cambridge, UK) and digitally captured by using the software GeneSys, version 1.5.7.0 (Syngene, Cambridge, UK). Strains with the same DGGE profile were grouped, and 2 representatives per group were amplified using the primers P1 and P4 targeting 700 bp of the V1–V3 region of the 16S rRNA gene (rDNA). Amplifications were carried out as reported above. After purification, amplicons were sent to a commercial facility for sequencing (Eurofins AG, Ebersberg, Germany). The sequences were aligned in GenBank using the Blast program, version 2.13.0 [36] to determine the closest known relatives of the partial 16S rDNA sequence obtained.

From the MEA plates, at day 0 of storage, only 20 colonies were isolated and transferred into the following three different medium cultures: Czapek Dox agar (Oxoid, Milan, Italy), MEA, and malt salt agar (5% malt extract; 5% NaCl; pH 6.2; Oxoid, Milan, Italy). The isolated moulds were identified according to the traditional methods proposed by Samson et al. [37] by examination of macroscopic (colonial) and microscopic characteristics of the moulds grown on the three agars. In particular, the morphology, shape, size, and colour of the colonies and spores, production of pigment, and type, development, and texture of the mycelium were observed. The presumptive identification was then confirmed by sequencing a fragment of approximately 600 bp of the D1–D2 region of the large-subunit rRNA gene according to the methods reported by Iacumin et al. [35]. Primers NL1 (5′-GCCATATCAATAAGCGGAAAAG-3′) and NL4 (5′-GGTCCGTGTTTCAAG ACGG-3′) were used. The reaction mixture and amplification protocol were the same as those described above. Sequence comparisons were performed in GenBank using the Blast program, version 2.13.0.

### 2.4. Physico-Chemical Analysis

Fifteen samples for each lot were collected and stored at the above temperatures. At each established time, three samples were subjected to physico-chemical analyses. The pH was measured at 3 different points using a pH metre (Crison Basic 20, Crison Instruments, Barcelona, Spain) by inserting the probe directly into the product. The water activity (Aw) was measured with an Aqua Lab 4 TE (Decagon Devices, Pullman, WA, USA). The humidity was measured according to the A.O.A.C [38], and NaCl and TVB-N (total volatile basic nitrogen) were measured according to Pearson [39].

Ethanol and lactic acid concentrations were determined in triplicate using the test ethanol (LOD: 0.093 mg/kg) and lactic acid (LOD: 0.21 mg/kg) from Neogen (Ayr, Scotland, UK).

Water phase salt was determined according to Huss et al. [40] using the following formula:$$\mathrm{WPS}\;(g/100\;\mathrm{mL})=\frac{salt\;content\;(in\;g\;per\;100\;g)}{moisture\;content\;(in\;mL\;per\;100\;g)\;+\;salt\;content\;(in\;g\;per\;100\;g)}\times 100$$

Thiobarbituric acid reactive substances (TBARSs) were determined according to Ke et al. [41].

### 2.5. Proximate Analyses

Fish fillets (500 g) were homogenized using a Grindomix GM200 knife mill (Retch, Haan, Germany) to obtain a homogeneous sample. Determination of water was performed by the gravimetric method (ISO, [42]) with the use of a Memmert UF75 Plus thermostat (Schwabach, Germany). The total protein content was determined using the Kjeldahl method (HRN ISO, [43]) with the use of the block for the destruction (Unit 8 Basic, Foss, Höganäs, Sweden) and the automated device for the distillation and titration (Kjeltec 8400, Foss, Höganäs, Sweden). The total fat was determined using Soxhlet (HRN ISO, [44]) by digestion of the sample by acid hydrolysis, followed by extraction of the fats by means of petroleum ether on a Soxtherm 2000 Automatic (Gerhardt, Munich, Germany). The ash content was obtained according to ISO [45] and with the use of furnace program controller LV 9/11/P320 (Nabertherm, Germany). The carbohydrate content was determined by calculation based on the determination of water, ash, total protein, and fat content. The mean value of the data was obtained from two parallel runs and expressed in g of component per 100 g of sample, with an accuracy of 0.01 g/100 g.

#### 2.5.1. Fatty Acid Profile

The sample preparation method for the analysis of fatty acid methyl esters was described by Pleadin et al. [46,47]. Methyl esters of fatty acids were analysed using gas chromatography (GC) according to the EN ISO [48] and EN ISO [49]. To the above effect, a 7890BA gas chromatograph equipped with a flame ionization detector (FID) and a 60 m DB-23 capillary column with an internal capillary diameter of 0.25 mm and a stationary phase thickness of 0.25 μm (Agilent Technologies, Santa Clara, CA, USA) were used. The components were detected by FID at a temperature of 280 °C, with a hydrogen flow of 40 mL/min, air flow of 450 mL/min, and nitrogen flow of 25 mL/min. The initial column temperature was 130 °C; after one minute, it was increased by 6.5 °C/min until a temperature of 170 °C was reached. The temperature was further increased by 2.75 °C/min until a temperature of 215 °C was attained. The latter temperature was maintained for 12 min and then further increased at a rate of 40 °C/min until the final column temperature of 230 °C was reached, and the latter was maintained for 3 min. One millilitre of a sample was injected into a split-splitless injector at 270 °C with a partition coefficient of 1:50. The carrier gas was helium (99.9999%), flowing at a constant rate of 43 cm/s. Fatty acid methyl esters were identified by comparing their retention times with those of fatty acid methyl esters contained in the standard mixture, as described earlier by Pleadin et al. [50]. The conversion of fatty acid methyl esters to fatty acid data per 100 g of fish muscle was performed according to the FAO/INFOODS Guidelines for Converting Units, Denominators, and Expressions [51]. For quality control purposes, reference material of fish oil T14206QC (Fapas, York, UK) was used. The results were expressed as a percentage (%) of a particular fatty acid in the total fatty acids, with an accuracy of 0.01%.

#### 2.5.2. Mineral Composition

After acid digestion of a 0.2 g homogenized sample with 7 mL of 60% nitric acid and 1 mL of 30% hydrogen peroxide (Perdrogen) in the microwave (Ethos Easy, Milestone, Italy), the sample was transferred in measuring flasks (25 mL), supplemented with water to the mark, and further diluted depending on the mineral. Mineral determination of minerals (mg/kg) was performed for sodium (Na), calcium (Ca), potassium (K), magnesium (Mg), copper (Cu), zinc (Zn), and iron (Fe). Analyses were performed by flame atomic absorption spectroscopy (200 Series A4 with SPS 4 Autosampler, Agilent Technologies, USA) at ʎ = 589.0 nm for Na, ʎ = 422.7 nm for Ca, ʎ = 766.5 nm for K, ʎ = 285.2 nm for Mg, ʎ = 324.8 nm for Cu, ʎ = 213.9 nm for Zn, and ʎ = 248.3 nm for Fe. For all minerals, HC coded lamps (Agilent Technologies, USA), specific for each mineral and its standard solution of 1000 μg/mL in 5% nitric acid, were used. The phosphorus content was determined according to standard ISO [52] by use of a spectrophotometer DR/4000U (Hach, Düsseldorf, Germany). To check quality control, reference material of fish muscle BB422, European Reference Materials, Institute for Reference Materials and Measurements (Geel, Belgium), was used. All solvents used for sample preparation and instrumental analyses were of HPLC grade. Ultrapure water was supplied by the Merck system Direct-Q3 UV (Kenilworth, New Jersey, USA). The mean of data obtained from two parallel runs was expressed as mg of mineral per kg of fish fillet, with an accuracy of 0.01 mg/kg.

#### 2.5.3. Vitamins A and E Content

Two grams of homogenized sample was weighed in a 50 mL amber PTFE tube. Then, 0.4 g of ascorbic acid was added in addition to 10 mL of a mixture of hexane/isopropanol (3:2). The samples were then vortexed for 60 s and extracted for 30 min on a head-over-head shaker. After extraction, the samples were centrifuged for 5 min at room temperature and 5000 rpm. The supernatant was transferred to a PP tube, 1 mL of 5% potassium hydroxide was added, and the samples were immediately vortexed for 60 s. Saponification was performed for 15 min in a water bath heated to 85 °C. Samples were then cooled to room temperature, and 10 mL of ultrapure water was added. Afterwards, the samples were extracted twice with 5 mL of hexane by means of gentle head-over-head shaking. The obtained supernatants were pooled and evaporated under a stream of nitrogen at 55 °C. After evaporation, the samples were reconstituted in 1 mL of ethanol and filtered through a 0.2 µm PTFE filter into vials. The instrumentation consisted of a UHPLC (degasser, pump, autosampler, and column compartment, 1290 Infinity) coupled with a UV detector, all delivered by Agilent Technologies (Santa Clara, CA, USA). Chromatographic separation was achieved on a Poroshell 120-C18 column, 150 × 3 mm, 2.7 µm (Agilent Technologies, Santa Clara, CA, USA). The mobile phase consisted of water (A) and methanol (B). A gradient elution was employed as follows: 0–2.5 min 15% A, 2.5–8 min 0% A, 8–10 min 15% A, with a flow rate of 2 mL/min and a column temperature of 40 °C. The injection volume was 15 µL, and the run time was 13 min. Both retinol and α-tocopherol were monitored by UV detection at λ_max_ = 295 nm. Analytical standard vitamin E (α-tocopherol, CAS 10191-41-0) was purchased from Supelco (Bellefonte, PA, USA), and vitamin A (retinol) (CAS 68-26-8) was purchased from Cerilliant (Austin, TX, USA). All solvents used for sample preparation and instrumental analyses were of HPLC grade. Ultrapure water was supplied by the Merck system Direct-Q3 UV (Merck Millipore, Darmstadt, Germany). The results of vitamin E content were expressed in mg/100 g to two decimal places and of vitamin A in µg/100 g to one decimal place.

### 2.6. Statistical Analysis

The data were analysed using Statistica 7.0, version 8 software (StatSoft, Tulsa, OK, USA). The values of the different parameters were compared by one-way analysis of variance, and the means were then compared using Tukey’s honest significance test. Differences were considered significant at *p* < 0.05.

## 3. Results and Discussion

### 3.1. Microbial Characteristics

The results for CSSB vacuum packing are presented in Table 1. The mean total bacterial count (TBC) at the beginning of storage reached 2.5 CFU/g, and at the end of the shelf life (60 days) reached 7.2 CFU/g. The final amount is lower than the level observed by different authors in CSS [12]. The level of TBC in CSS could reach 8–9 log CFU/g; however, at this concentration, no sensory defects were detected, and consequently, TBC cannot be used as a sensory indicator for CSS [15,53].

In Italy, there is no TBC limit for CSF. However, the limit proposed by Sernapesca [54] and by Spain Ministerio de Sanidad y Consumo [55] could be used. They established limits of the TBC between 5 or 6 log CFU/g for CSF, but this limit is easily exceeded, considering the long shelf life and the temperature abuse observed in the family or supermarket fridge, which is always at a level of 7–7.5 °C. Indeed, Truelstrup Hansen et al. [19] also found CSS with a high TBC (8 log CFU/g), but they concluded that the products were not spoiled. Similar results were obtained by Dondero et al. [22] when storing vacuum-packed CSS at various temperatures (from 0 to 8 °C) and demonstrating a correlation of the TBC with the storage time. The level of contamination observed in our study throughout the storage time could be due to the temperature abuse, as demonstrated by the storage test method. In fact, the final contamination level of the TBC of the sample stored at 4 °C was lower than 5 log CFU/g (data not shown). However, it is well recognized that the level of TBC at the end of the shelf life of CSF is always more than 6 log CFU/g, independent of the type of packaging (MAP, VP, or air) and the storage temperature [12,27].

The low levels of TBC at the beginning of storage indicate good fish quality [4,12,56], which could be attributed to the raw material, packaging material, and processing, as demonstrated by Truelstrup Hansen et al. [57], who studied the influence of three different smokehouses on the contamination level of vacuum-packed CSS and cold smoked trout (CST). The results showed a significant difference in microbial counts among the three different Danish smokehouses. Recently, Wiernasz et al. [17] obtained similar data in a study aimed at investigating the influence of the manufacturing process on the microbiota of CSF, demonstrating that the processing and packaging conditions affected the microbial composition and the quality of the final product. Other authors also demonstrated that aerobic plate counts (APCs), psychrotrophic clostridia, LAB, presumptive *Aeromonas*, and presumptive staphylococci in CSS were not influenced by the smokehouse [21].

The initial Enterobacteriaceae count was less than the detection limit (<10 CFU/g), confirming previous data obtained by Bernardi et al. [53] in CSS (Table 1). Subsequently, it increased until the end of the storage and reached up to 3 log CFU/g, demonstrating a significant difference between the initial and final contamination (*p* < 0.05). Regarding CSF, the Enterobacteriaceae counts are variable but not always significantly different (*p* > 0.05) [21]. The initial level of contamination seemed to be related to the in-house microbiota and hygienic conditions in the smokehouses rather than to the raw material quality, which always reaches up to 100 CFU/g for fresh fish meat [4,53,58].

The level of Enterobacteriaceae in CSF and CSSB at the end of their shelf life is always more than 6–7 log CFU/g, independent of the initial contamination [4,59]. During storage, psychrotrophic Enterobacteriaceae can grow and reach high loads, as demonstrated by Fuentes et al. [59]. The presence of Enterobacteriaceae can also be associated with the spices used in the production of CSF, as they grow preferably in CSS supplemented with dill [17].

Typically, the Enterobacteriaceae load is very low after production, but it increases during storage depending on the type of salting and packaging used, even if MAP and VP show very low and closed counts [59]. Our results agree with those of other authors who reported that MAP increases the shelf life of fish products with respect to those packaged in air; however, it confers little or no additional shelf life when compared with VP [59].

The concentration level of Enterobacteriaceae can be an index of spoilage. Lightly preserved fish products may contain high levels of Enterobacteriaceae, which are often associated with smoked fish deterioration [15,21]. Only Dondero et al. [22] found a low total coliform count of less than 43 MPN/g in CSS stored at 8 °C. Sernapesca [54] proposed too restrictive limits for Enterobacteriaceae (total coliforms and *Escherichia coli*) in CSS, which can be accepted only when the total coliform and *E. coli* counts are less than 10 MPN/g and 4 MPN/g, respectively. Considering the different Enterobacteriaceae method of analysis, it was impossible to use the Sernapesca [54] limits to consider the acceptability of our investigated samples.

The LAB count at the beginning of storage was 2.5 CFU/g and corresponded to the TBC count. During storage, the LAB concentration increased, and on day 30, LAB became predominant. Their predominance over the other microbial populations increased, and at the end of shelf life, the LAB count reached 7.6 CFU/g. As observed also by other authors, LAB were dominant [20].

In particular, high levels of LAB (7–8 log CFU/g) have been observed in CSF for several weeks before the product is discarded due to sensory defects, which clearly shows that sensory indicators cannot be used as a sign of the total bacteria count for CSS [12,15,53].

The dominance of these bacteria in vacuum-packed foods, such as CSF, can be explained by their ability to grow rapidly under anaerobic conditions and at low temperatures and their tolerance to CO_2_ [20,22]. LAB are known to produce organic acids, mainly lactic acid and ethanol, as fermentation end products [60], but they can also produce nitrogen and sulfur compounds responsible for spoilage [61]. Despite their predominance in CSF, the roles of LAB in spoilage are not as clear as many authors have shown; in fact, there is no correlation between shelf life and LAB count or any other bacterial number [25]. A high LAB load is not often correlated with spoilage [57]. According to Dondero et al. [22], *Lactobacillus* spp. showed a correlation with storage time and sensory quality at all storage temperatures. Moreover, Bernardi et al. [53] observed a huge variability in the LAB concentration in their investigated samples and, consequently, hypothesized that LAB, like other bacterial groups, have been identified as weak or strong spoilage organisms, depending on the strain.

The present study supports the findings of other authors, which have highlighted a substantial variation in bacterial numbers among CSS in retail [53,62].

Yeasts and moulds were also quantified in the studied CSSB. Both microorganisms were present at a level of 10 CFU/g and did not change from the beginning to the end of storage. Moulds are typically aerobes, and, therefore, they cannot grow in VP products lacking oxygen. Yeasts can be either aerobes or fermentative (anaerobes), and their growth is influenced by temperature and food composition. In the investigated CSSB, only *Debaryomyces hansenii* was isolated (Table 2), although it preferably grows in aerobic ecosystems. However, moulds and yeasts have often been isolated from spoiled CSF [20], and their presence indicates a relatively complex and variable microbiota in this type of product. Staphylococci coagulase-positive, sulfite-reducing *Clostridium*, *Salmonella* spp., and *L. monocytogenes* were not isolated from any of the analysed samples. This demonstrated the good quality of the raw fish and the smoking process [63]. The physico-chemical parameters of cold smoked sea bass stored at 4 ± 2 °C for 2 days and then at 8 °C until 60 days are shown in Table 3.

No significant differences were found between the pH values observed throughout the storage period (*p* > 0.05). The pH values ranged from 5.94 to 5.99 and were lower than those observed in other CSF, which were 6.12 for Italian CSS [53] and 6.20 for French CFS [58]. Small pH differences were observed in Spanish CSS (5.65–6.09) and in Spanish CST (5.71–6.11) [21].

The observed differences could be due to the type of raw meat studied (sea bass vs. salmon) and to the production process. It is well known that the residual glycogen concentration of fish meat after slaughtering is lower than that of the meat of terrestrial animals [64], and, therefore, anaerobic glycolysis, which produces lactic acids, is limited, resulting in a higher pH. This is further confirmed by the lactic acid value observed during the entire storage period (Table 1).

The Aw remained within the range of 0.970 to 0.971. In all the samples, no significant difference was observed during storage (*p* > 0.05). The level of Aw is higher than those observed by other authors in CSS samples. Bernardi et al. [53] did not observe any Aw variation during CSS storage, and the Aw values ranged from 0.916 to 0.956. An Aw level of approximately 0.95 was observed in Spanish CSS and CST by Gonzàlez-Rodrìguez et al. [21]. The Aw difference, however, is demonstrated by a lower salt concentration (3.2 vs. 3.43%) observed in our samples compared to what was observed in Italian and French CSS [53,58]. The Aw values of the investigated samples are similar to those observed for other CSF. Usually, the Aw in CSS is between 0.96 and 0.98 [12,65].

The moisture content remained constant over time, as demonstrated in a previous work [4]. It exhibited values between 59.18% and 59.25%, and the observed differences were due to the different samples analysed rather than the absorption or loss in moisture (Table 3).

Our results were compared with those of Leroi et al. [58] on thirteen French commercial products intended to be representative of French traditional production and those of Bernardi et al. [53]. The values for moisture content (approximately 59.2%) obtained in our study were not significantly different (*p* > 0.05) from the French traditional products (60.5%) but were significantly different from those of Bernardi et al. (66.3%) [53].

The salt content observed was 3.2% and was similar to that found by Leroi et al. (3.13%) [58] but different from that found by Bernardi et al. (3.43%) [53]. Espe et al. [62] reported data that was different from ours about chemical–physical parameter monitoring in a study of 48 French commercial CSS. They found a salt level of 2.62% and a water content of 62.5%. Finally, Cornu et al. [66] reported these data from monitoring of 40 French commercial products: salt content 2.85%, moisture 61.3%, and WPS 4.62%, and Gonzàlez-Rodrìguez et al. [21] reported for Spanish CSF a level of WPS between 4.30 and 4.75% in CSS and between 4.11% and 4.70% in CST.

Comparing our results to those of the aforementioned authors, it can be hypothesized that the differences are likely due to the salting and smoking technologies used. Indeed, it is well recognized that the duration of salting and smoking can influence the physico-chemical parameters of smoked products.

In this study, the WPS value varied according to the tested samples, and in each case, it was over 5.0% (Table 3). These values are acceptable for CSF [4]. In this study, the salt and WPS values varied over the storage period without showing a specific trend or significant differences, indicating that the observed differences were due only to variability among the samples. Most likely, the salt content was influenced by the variability in the samples and the salting procedure. For these reasons, the observed decreases cannot be considered trends but were due to random heterogeneity in the samples.

According to the data obtained from the aforementioned French and Italian authors, the mean salt and moisture contents were low. Indeed, the WPS mean in the CSSB object of our work was higher (5.2 vs. 4.93/4.62%) than the French, Italian, and Spanish products studied by various authors [24,53,58,62,66]. High WPS values in CSF are recommended because it is the only parameter that can inhibit the growth of psychrotrophic *C. botulinum*, considering the type of packaging (without oxygen) and the long shelf life of the CSF. The Center for Food Safety and Applied Nutrition [67] recommended a WPS level over 3.5%, which, combined with a refrigerated storage temperature (<4.4 °C), allows the control of spore germination of psychrotrophic *C. botulinum*. In the current study, all the tested samples had a WPS value greater than 5.0%, consequently they did not support the growth of *C. botulinum*. Despite the presence of lower WPS in CSF in the three different areas mentioned above, none of the samples supported *C. botulinum* growth.

### 3.2. TVB-N Characteristics

The average values of TVB-N and TBARSs (Table 3) were largely acceptable. The CSSB showed TVB-N values of 28.2 mg N/100 g immediately after packaging, which evolved to 34.3 mg N/100 g at the end of the shelf life. Significant increases were observed over time, which was limited to 3–5 mg N/100 g (*p* < 0.05). Based on the TVB-N data, the products can be accepted at day 60, considering that they were lower than the limit of 40 mg N/100 g proposed for CSS by Cantoni et al. [68]. However, the level of TVB-N exceeded the restrictive limit according to the Chilean fishing authority [54], which was established at 30 mg TVB-N/100 g. To date, an EC regulation about the value of TVB-N does not exist, and consequently, various authors usually apply limits based on their experience or on the correlation among the microbial and TVB-N loads and the sensory analysis [22,53,58]. The limit proposed by Cantoni et al. [68] seems to be more realistic, and the end of the shelf life of an SSF can be acceptable when the TVB-N exceeds 40 mg N/100 g [53].

Dondero et al. [22] found an increase in TVB-N in CSS stored at different temperatures (0, 2, 4, 6, and 8 °C). They observed that TVB-N increased with increasing temperature, and the total aerobic and anaerobic populations showed a positive correlation with TVB-N at all temperatures.

Levels from 15.5 to 52.8 mg TVB/100 g, with a maximum production rate during the last 2 weeks, were found by Leroi et al. [69] in CSS at 8 °C. In our study, the increase started after 15 days of storage, and the difference in TVB-N concentration between the start and the end was approximately 5.3 mg N/100 g, demonstrating light microbial and enzymatic spoilage.

The increase in TVB-N is usually caused by a combination of microbiological and autolytic deamination of amino acids and the complete microbial reduction of TMAO to TMA [19,22,64]. In particular, it seems that the higher producers of TVB-N in CSF are Enterobacteriaceae*, Photobacterium* spp., and *Lactobacillus* spp. [69], which are microorganisms that were present and grew in the CSSB samples investigated in our work (Table 2).

The TBARSs increased during storage; however, their level was lower than 6.4 nmol malonaldehyde/g at the end of storage (60 days). However, there was an increase in this parameter over time. The TBARS values were 5.5 and 5.6 nmol malonaldehyde/g at day 0 and day 15, respectively, demonstrating no significant difference (*p* > 0.05). Then, the TBARS values increased significantly, reaching levels of 6.2 nmol malonaldehyde/g at 30 days, 6.3 nmol malonaldehyde/g at 45 days, and 6.4 nmol malonaldehyde/g at 60 days (*p* < 0.05). The observed TBARS values demonstrated that the CSSB samples of this study can be acceptable and are not rancid. Fish products with TBARS values lower than 8 nmol/g are considered not rancid, those lower than 9–20 nmol/g are slightly rancid but still acceptable, and those more than 21 nmol/g are rancid and unacceptable [41,70]. Bernardi et al. [53] examined CSS sold at retail outlets in Italy and concluded that five CSS samples were not rancid, seven samples were still acceptable, and two were unacceptable. Values above 8 nmol/g are frequent [62], depending on salting and smoking methods. The mean values of TBARSs in our samples were lower, similar to the results obtained by Espe et al. [62], who found values of 8.7 nmol/g, 8.1 nmol/g, and 7.8 nmol/g for Norwegian, Scottish, and Irish smoked salmon, respectively.

### 3.3. Ethanol and Lactic Acid Presence

The mean values of ethanol and lactic acid are shown in Table 3. The values of these compounds increased from the start to the end of the shelf life. The final concentrations on the last day of storage reached 186.5 mg/kg and 201.3 mg/kg for ethanol and lactic acid, respectively. The concentrations of both compounds were significantly different (*p* < 0.05) from the beginning to the end of storage. Their presence originates from anaerobic glycolysis after slaughtering, smoking, and LAB growth, both homo- and heterofermentative [64]. The low amount of these compounds is due to the relatively low concentration of sugars in fresh fish meat [64]. Gonzàlez-Rodrìguez et al. [21] found that the ethanol concentration depended on the types of fish. Indeed, they found that in salmon fillets, the ethanol concentration was significantly higher than that in trout fillets. In a previous study, no significant negative correlation (*p* > 0.05) was observed between ethanol levels and salt content [19]. The presence of ethanol or lactic acid cannot be considered a quality index for CSF because, as shown in different studies, the levels of ethanol did not correlate with bacterial counts and sensory analysis [21].

### 3.4. Types of Microorganisms of CSSB

The investigated CSSB were contaminated by a large population of microorganisms, which originate from the raw material, the environment, and the processing, and their growth depends on different conditions (temperature, Aw, pH, microbial interactions), as shown for CSS in previous studies [12,64]. Table 2 shows all the strains isolated in the investigated CSSB. Among the 268 isolates, 31% were Gram-negative, 62% were Gram-positive, and 7% were fungi (yeasts and moulds). The number of isolates changed through the storage time, and 5 out of 18 isolated strains were always present at all the times of investigation. In particular, the predominant species included *P. phosphoreum*, *B. thermospacta*, *Lactiplantibacillus plantarum*, *Latilactobacillus sakei*, and *D. hansenii*, which represented approximately 56% of the total isolates. Except for *D. hansenii*, their presence and growth are demonstrated by the ability to live in anaerobic ecosystems [12,21,64], which is confirmed by the increase in the TBC and LAB through the CSSB storage time (Table 1). Considering the number of isolated microorganisms, it was confirmed that in this type of CSF, LAB predominates, but *P. phosphoreum* and *B. thermosphacta* were also isolated; these strains are frequently isolated and represent the typical cold-tolerant microorganisms of refrigerated VP, CSS, and CSF [12,21,53,58,64]. Other LAB were also isolated and included *Carnobacterium maltaromaticum*, *Latilactobacillus curvatus*, and *Loigolactobacillus coryneformis*. Their presence was sporadic, and except for *C. piscicola*, their isolation was limited to the day of sampling. These species were often isolated in CSS by several authors, who never associated them with spoilage [12,64]. Gonzàlez-Rodrìguez et al. [21] isolated *C. piscicola* in CSS and CST and showed that it predominated over the other LAB.

The LAB strains present in our samples were different than those isolated by other authors. Among LAB, the dominance of some species over others is strictly dependent on the type of flesh, the smokehouse, and the processing area [18,20]. Enterobacteriaceae belonging to the species *Hafnia alvei*, *Serratia liquefaciens*, and *Proteus vulgaris* were also isolated in the investigated CSSB, but their number was limited, representing only 10% of the total isolates. According to Gonzàlez-Rodrìguez et al. [21], Enterobacteriaceae can constitute the second largest group present in CSF, and the dominant genera were *Proteus* and *Serratia*, which were also identified in our samples.

*Aeromonas* and *Staphylococcus xylosus* were also isolated in this study, although their presence was sporadic and limited. Aerobic Gram-negative bacteria (*Moraxella*, *Pantoea agglomerans*, and *Pseudomonas fluorescens*) and Gram-positive bacteria (*Bacillus licheniformis*) represented approximately 13% of the isolates and must be considered sporadic contamination [15,19,20,29]. Yeast and moulds were also found in the investigated CSSB, but only two species were isolated: *D. hansenii*, which was detected all through the period of storage; and *Penicillium nalgiovense*, which was isolated only on day 0. Their presence is probably linked to the salting step or accidental contamination. *Debaryomyces hansenii* is, in fact, a typical yeast of salted and ripened meat, and *P. nalgiovense* is a typical mould used as a starter for sausages and, therefore, can contaminate the ripening area [64]. The laboratory produces either fish or meat products. Consequently, this could probably be the reason for the presence of the two aforementioned microorganisms, typically linked to sausage production, in CSSB.

The microbiota of refrigerated vacuum-packed CSF appears to be highly variable and related to the raw materials, producing environments, and processing [21]. Studies from which the relative incidence of LAB genera can be determined show great variation in vacuum-packed CSF, although overall, the genera and species found in our investigated CSF have been reported in previous studies [12,20,21,53,58,64]. Although spoilage of meat under anaerobic conditions is attributed to LAB [23], there is considerable confusion over the role of this group in vacuum-packed CSF. However, *Carnobacterium* and *Lactobacillus* species [20] have demonstrated their potential activity in CSF spoilage in vitro, but it was suggested that autolytic enzymes can also contribute to product deterioration. *L. sakei*, *L. farciminis*, and *B. thermosphacta* produced sulfurous and acidic off-odours [71] in sterile cold smoked salmon stored in vacuum packs at 6 °C, and some isolates of *P. phosphoreum* were characterized by an acidic effect [71].

Based on these considerations and the results of this study, it could be hypothesized that the spoilage of CSSB is due to the combination of LAB, *P. phosphoreum,* and *B. thermosphacta* activity. Their activity is demonstrated by the increase in their concentration and typical spoilage metabolites, such as TVB-N, ethanol, and lactic acid, during storage. These metabolites were found in the investigated CSSB samples and were also found in other studies [12,20,21,53,58,64].

### 3.5. Nutritional Characteristics

Analysis of basic nutritional parameters showed that CSSB are a valuable source of protein (23.7%), while water and fat account for approximately 80% of the composition. Due to the high water content (60%) and low fat content (11.6%), it can be considered a low calorie food (200 kcal/100 g) (Table 4). Based on the nutritional composition, it can be assumed that this type of product meets the expectations of modern consumers. Regarding consumer preferences for fishery products, research shows that today they are looking for a healthy, readily available product that can be prepared quickly [72].

The fatty acid profile generally showed a dominance of the two classes of monounsaturated fatty acids (MUFAs) and polyunsaturated fatty acids (PUFAs), while the proportion of saturated fatty acids (SFAs) was the lowest (Table 5). The ratio of PUFAs to SFAs was approximately 1.5, which should be considered adequate since, according to recommendations, a ratio of PUFAs to SFAs higher than 0.4 reduces the risk of autoimmune, cardiovascular, and other chronic diseases [73]. In the studied fish fillets, the content of n-6 fatty acids was almost twice that of n-3 fatty acids (Table 5), and the obtained value of the n-6/n-3 ratio of 1.9 is considered advisable from a nutritional point of view. A ratio of n-6/n-3, which should be lower than 4, could reduce the incidence of diet-related diseases [73,74]. It is well known that the fatty acid profile of farmed fish reflects the content and fatty acid profile of lipid sources in the diet. Consequently, the choice of dietary lipids would lead to a tailored fatty acid composition of farmed fish to incorporate beneficial health aspects according to consumer expectations.

In the MUFA fraction, oleic acid (OA, C18:1 n-9) was predominant (36–68% of total fatty acids, TFA) (Table 6). Linoleic acid (LA, C18:2 n-6) was the major PUFA (18.7%TFA), followed by alpha-linolenic acid (ALA, C18:3 n-3), docosahexaenoic acid (DHA, C22:6 n-3), and eicosapentaenoic acid (EPA, C20:5 n-3). The predominance of n-6 PUFAs has been confirmed in other studies on farmed sea bass [75,76]. Linoleic acid is derived from vegetable oils used in the production of fish feed, and, because fish have a lower capacity for chain elongation and desaturation, this fatty acid is enriched in unmodified form in the lipids of marine fish [77]. Fatty acids with C20 and C22 are more valuable than C18 from a nutritional point of view, as the greatest importance is attributed to EPA and DHA, which are largely responsible for a favourable n-6/n-3 ratio in fish [50]. As reviewed by Tilami and Samples [78], these fatty acids are essential in the human diet and have been shown to be involved in many metabolic functions. They have immunosuppressive properties and exhibit anti-inflammatory effects, reduce platelet aggregation, build cell membranes, and significantly improve cardiovascular, nervous, and brain tissue function. The EFSA (European Food Safety Authority) has suggested a daily intake of 250 mg EPA + DHA, 2 g ALA, and 10 g LA [79]. Consumption of 1 to 2 servings of fish per week could have a protective effect against ischaemic stroke and coronary heart disease [80]. Palmitic acid (PA C16:0) was the most important SFA, accounting for 14.29% of the total fatty acids (Table 6). The results for fatty acid composition obtained in this study can be considered similar to those determined for fresh farmed sea bass samples cultivated in different parts of the Mediterranean region [56,75,81,82].

Although fish is generally considered a good source of minerals such as potassium, phosphorus, iodine, sodium, calcium, and vitamins D, A, and E [78], it is known that the mineral composition of the same fish species can vary greatly. The variations in the mineral composition of fish flesh are closely related to the food source, environmental conditions and area of catch, size, age, sexual maturity, etc. [75,81]. The contents (mg/100 g) of the studied minerals, which can be classified as macroelements (phosphorus (P), potassium (K), sodium (Na), calcium (Ca) and magnesium (Mg)) and microelements (iron (Fe), zinc (Zn) and copper (Cu)), are summarized in Table 7.

The content of selected minerals in CSSB decreased in this study as follows: Na > K > P > Mg > Ca > Fe > Zn > Cu. For macroelements, the concentration ranged from 317.23 mg/kg (calcium) to 15,744.67 mg/kg (sodium), while for microelements, the analysis showed the highest concentration for iron (23.69 mg/kg) and the lowest concentration for copper (0.87 mg/kg) (Table 7). Smoked sea bass, due to its favourable mineral content, represents a good source of P, K, Ca, Mg, and Fe (Regulation EC, [83]). Compared to the results of other studies that addressed the mineral composition of farmed sea bass, the sea bass samples from this study contained more sodium but less calcium and less phosphorus [56,75,76,81,82,83,84]. The main functions of these essential minerals include building the skeleton, maintaining the colloidal system, and regulating the acid–base balance, as well as building enzymes and their activators and hormones [85].

The potassium concentration (5155.67 mg/kg) was similar to the values reported by Erkan and Özden (4597 mg/kg) [84] and Özden and Erkan (4601 mg/kg) [82] but lower than the 6287.5 mg/kg obtained by Kocatepe and Turan [81]. Adequate intake of potassium, one of the three electrolytes, is beneficial for health. It is the main positively charged electrolyte in cells and has the role of maintaining fluid balance and acid–base balance. It is also important for maintaining heart rhythm, muscle contraction, transmission of nerve impulses, and protein synthesis [86].

Magnesium is an essential nutrient involved in virtually all major metabolic and biochemical processes in the cell. The magnesium concentration observed (577 mg/kg) was higher than that in Erkan and Özden (326 mg/kg) [84], Özden and Erkan (325 mg/kg) [82], Kocatepe and Turan (380.5 mg/kg) [81] and Fuentes et al. (0.11 mg/g) [56] but lower than that in Bhouri et al. (1740 mg/kg) [76] (Table 7). Iron, zinc, and copper are the most abundant microelements in fish and have many biological functions [56,75]. The iron concentration (23.69 mg/kg) was similar to the values determined by Erkan and Özden [84] and Özden and Erkan [82], lower than the value of 51.2 mg/kg determined by Alasalvar et al. [75], but higher than the data obtained by other authors [56,76,81,87]. Because iron is essential for the acquisition, distribution, storage, and use of oxygen, symptoms of deficiency include anaemia and impaired physical performance, psychomotor development and function, and immune function [88].

Zinc is a mineral that is a component of numerous enzymes and is involved in DNA synthesis, growth, development, and immune system functioning. Zinc and copper are carefully regulated in humans by physiological mechanisms. Although they can be toxic if consumed in excess, the concentrations found in farmed sea bass fillets are below the level that could be harmful to health. The samples studied were a better source of zinc (13.36 mg/kg) compared to the samples studied by Erkan and Özden [84] and Özden and Erkan [82], which had values of 2.8 mg/kg, and even better than the 8.4 mg/kg found by Kocatepe and Turan [81] or Yildiz et al. [87] with a value of 3.6 mg/kg. Only Bhouri et al. [76] and Alasalvar et al. [75] obtained much higher values (53.4 mg/kg and 45.1 mg/kg, respectively). The copper concentration was quite low (0.87 mg/kg), and was lower than previously reported values of 3.7 mg/kg [76] and 3.87 mg/kg [75] but higher than the results obtained by other authors [56,81,87].

In this study, two vitamins were analysed in cold-smoked fish fillets, vitamin A and vitamin E, among which only vitamin E was detected (>LOD) (Table 8). The main benefit of vitamin A and vitamin E is their antioxidant activity. The determined concentration of vitamin E was in accordance with the values determined in sea bass by other authors [89,90]. Both of the mentioned vitamins belong to the class of fat-soluble vitamins. For this reason, vitamin A is more abundant in fish oil and liver than in fish fillets and was detected at levels below 0.5 μg/100 g.

## 4. Conclusions

Sea bass is a typical product of aquaculture, which represents one of the major food production systems that has widely grown over the last decades due to the increasing demand for seafood, which includes fresh or processed meat. Cold smoked sea bass is a processed food product of Italian fishery. It is an important source of high-quality protein for humans, and there is increased awareness of the beneficial effects of fish consumption on human health. Indeed, CSSB offers a source of macro- and micronutrients essential for the proper functioning of the human body. It is also rich in protein and omega-3 fatty acids. However, due to its high Aw, neutral pH, relatively high contents of nitrogen compounds and free amino acids, and presence of autolytic enzymes, it is very susceptible to microbiological and chemical spoilage. Microbial and physico-chemical monitoring demonstrated that the CSSB investigated had a high nutritional and hygienic quality. Pathogenic microorganisms were never detected, and it was demonstrated that the product could have a shelf life of up to 60 days by analysing chemical–physical and microbiological parameters. The microbial evolution was adequate, and at the end of the shelf life, it was lower than 8 log CFU/g. The TBARS values were largely acceptable as well as the TVB-N, which remained below 35 mg N/100 g. It can be concluded that CSSB can be considered safe for its excellent microbial quality and a functional food because of the high nutritional value of essential proteins and lipids.

## Figures and Tables

**Table 1 foods-12-02685-t001:** Microbial evolution in cold smoked sea bass stored at 4 ± 2 °C for 20 days and then at 8 °C till 60 days.

Days	TBC	LAB	Enterobacteriaceae
0	2.5 ± 0.4 a	2.5 ± 0.2 a	<10
15	3.4 ± 0.6 b	2.3 ± 0.8 a	2.0 ± 0.1 a
30	3.8 ± 0.4 b	3.9 ± 1.3 a	2.8 ± 0.5 b
45	6.5 ± 0.9 c	6.9 ± 1.1 b	2.9 ± 0.1 b
60	7.2 ± 0.3 c	7.6 ± 0.8 b	3.1 ± 0.3 b

Legend: TBC: total bacteria count; LAB: lactic acid bacteria. All data are represented as mean ± standard deviation; means with same letters following the columns are not significantly different (*p* > 0.05).

**Table 2 foods-12-02685-t002:** Microbial species in smoked sea bass.

	Species	Accession Number	Number of Isolates	Days				
				0	15	30	45	60
Gram −	*Hafnia alvei*	NR_044729.2	13		2	3	5	3
	*Serratia liquefaciens*	NR_042062.1	13		2	5	4	2
	*Pantoea agglomerans*	NR_041978.1	13		2	4	5	2
	*Proteus mirabilis*	NZ_CP049755.1	2		2			
	*Pseudomonas fluorescens*	NR_043420.1	6		3	2	1	
	*Aeromonas* spp.	NR_043638.1	5		2	3		
	*Moraxella* spp.	LTKA01000005.1	13		4	5	4	3
	*Photobacterium phosphoreum*	AY435161.1	19	8	4	3	2	2
Gram +	*Brochothrix thermospacta*	AB680248.1	47	7	10	11	9	10
	*Staphylococcus xylosus*	NZ_CP008724.1	7	3	4			
	*Bacillus licheniformis*	NR_074923.1	2		2			
	*Carnobacterium maltaromaticum*	NR_044710.2	26	3	6	10	3	
	*Lactiplantibacillus plantarum*	AP018405.1	51	10	8	11	12	10
	*Latilactobacillus curvatus*	NR_042437.1	6		2		2	2
	*Latilactobacillus sakei*	LT907930.1	19	3	3	5	4	4
	*Loigolactobacillus coryniformis*	AJ418918	8	8				
Moulds	*Penicillium nalgiovense*	JQ434685.1	3	3				
Yeasts	*Debaryomyces hansenii*	NC_006047.2	15	3	3	3	3	3

**Table 3 foods-12-02685-t003:** Physico-chemical parameters in cold smoked sea bass stored at 4 ± 2 °C for 20 days and then at 8 ± 2 °C till 60 days.

Days					
	0	15	30	45	60
pH	5.94 ± 0.05 a	5.95 ± 0.03 a	5.96 ± 0.05 a	5.99 ± 0.05 a	5.95 ± 0.03 a
% Moisture	59.11 ± 0.12 a	59.18 ± 0.09 a	59.25 ± 0.12 a	59.20 ± 0.18 a	59.15 ± 0.24 a
% NaCl	3.2 ± 0.11 a	3.2 ± 0.18 a	3.1 ± 0.20 a	3.1 ± 0.18 a	3.1 ± 0.19 a
Aw	0.970 ± 0.002 a	0.970 ± 0.001 a	0.971 ± 0.005 a	0.971 ± 0.003 a	0.970 ± 0.005 a
% WPS	5.2 ± 0.1 a	5.1 ± 0.2 a	5.1 ± 0.3 a	5.2 ± 0.2 a	5.3 ± 0.2 a
TVB-N mg N/100 g	28.2 ± 0.8 a	29.0 ± 0.1 a	32.2 ± 0.3 b	33.20 ± 0.2 c	34.30 ± 0.2 d
TBARSs nmol/g	5.5 ± 0.2 a	5.6 ± 0.2 b	6.2 ± 0.3 c	6.3 ± 0.2 c	6.4 ± 0.1 c
Ethanol	9.2 ± 3.3 a	26.5 ± 6.8 b	54.2 ± 12.5 c	130.5 ± 9.3 d	186.5 ± 9.5 e
Lactic acid	10.3 ± 1.3 a	29.5 ± 8.8 b	60.2 ± 9.3 c	140.7 ± 10.8 d	201.3 ± 15.3 e

Legend: WPS: water salt phase; TVB-N: total volatile nitrogen; TBARSs: thiobarbituric acid reactive substances—malonaldehyde index; ethanol, mg/kg; lactic acid, mg/kg. Data represent the means ± standard deviations of the total samples; means with the same letters follow the lines and considering each parameter are not significantly different (*p* < 0.05).

**Table 4 foods-12-02685-t004:** Basic chemical composition of fish products.

Parameter	Smoked Fish Fillets
Energy (kJ/kcal)	834/200
Water (%)	60.0 ± 0.1
Protein (%)	23.7 ± 0.2
Fat (%)	11.6 ± 0.1
Ash (%)	5.1 ± 0.1
Carbohydrates (%)	<0.5

**Table 5 foods-12-02685-t005:** Fatty acid composition of fish products.

Fatty Acids (g/100 g)	Smoked Fish Fillet
SFA	2.3 ± 0.0
MUFA	5.0 ± 0.0
PUFA	3.5 ± 0.2
Total n-6	2.3 ± 0.1
Total n-3	1.2 ± 0.1
ALA	0.4 ± 0.0
EPA	0.3 ± 0.0
DHA	0.4 ± 0.0

**Table 6 foods-12-02685-t006:** Fatty acid profile (as % of total fatty acids).

Fatty Acids	Smoked Fish Fillet
C14:0	2.07 ± 0.05
C14:1	0.04 ± 0.00
C15:0	0.23 ± 0.01
C16:0	14.29 ± 0.36
C16:1 n-7*t*	0.24 ± 0.00
C16:1 n-7*c*	3.51 ± 0.04
C17:0	0.39 ± 0.01
C17:1	0.22 ± 0.01
C18:0	3.64 ± 0.10
C18:1 n-9*c*	36.68 ± 0.54
C18:1 n-7	2.91 ± 0.04
C18:2 n-6*t*	0.31 ± 0.01
C18:2 n-6*c*	18.70 ± 0.28
C18:3 n-6	0.29 ± 0.01
C18:3 n-3 (ALA)	3.74 ± 0.17
C18:4 n-3	0.40 ± 0.03
C20:0	0.24 ± 0.01
C20:1 n-9	2.47 ± 0.04
C20:2 n-6	0.77 ± 0.01
C20:3 n-6	0.14 ± 0.01
C20:4 n-6	0.27 ± 0.02
C20:3 n-3	0.21 ± 0.01
C20:4 n-3	0.29 ± 0.02
C20:5 n-3 (EPA)	2.21 ± 0.23
C22:0	0.12 ± 0.00
C22:1 n-11	0.79 ± 0.01
C22:1 n-9	0.28 ± 0.00
C22:2 n-6	<0.1
C23:0	0.24 ± 0.02
C22:5 n-3 (DPA)	0.15 ± 0.00
C24:0	0.70 ± 0.06
C22:6 n-3 (DHA)	3.25 ± 0.40
C24:1 n9	0.21 ± 0.00

**Table 7 foods-12-02685-t007:** Mineral composition of fish products.

Mineral	Smoked Fish Fillet
Phosphorus (P) (mg/kg)	2105.06 ± 10.11
Potassium (K) (mg/kg)	5155.67 ± 687.07
Sodium (Na) (mg/kg)	15,744.67 ± 356.57
Calcium (Ca) (mg/kg)	317.23 ± 46.67
Magnesium (Mg) (mg/kg)	577.77 ± 93.52
Iron (Fe) (mg/kg)	23.69 ± 4.49
Zinc (Zn) (mg/kg)	13.36 ± 0.88
Copper (Cu) (mg/kg)	0.87 ± 0.06

**Table 8 foods-12-02685-t008:** Vitamin A and E contents in fish products.

Vitamin	Smoked Fish Fillet
Vitamin A (µg/100 g)	<0.5
Vitamin E (mg/100 g)	0.47 ± 0.03

LOD (vitamin A) = 0.5 µg/100 g; LOD (vitamin E) = 0.1 mg/100 g.

## Data Availability

The data in this study are readily available upon reasonable request to the corresponding author.

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
