# Peer review of "Microbial and Physico-Chemical Characterization of Cold Smoked Sea Bass (Dicentrarchus labrax), a New Product of Fishery"

_foods, 2023, doi:10.3390/foods12142685_

Round 1

Reviewer 1 Report

The manuscript, ID foods-2462315, investigates microbial analysis, proximate contents, fatty acid profile, mineral composition, and vitamin contents for the purpose of identifying the microbiological and physicochemical characteristics of cold smoked sea bass, a new aquatic product. In my opinion, this study is considered to have useful value as a reference material for basic science on the nutritional components of cold smoked sea bass. Overall English writing was fine, and no issues detected. Also, reliable research results were provided based on systematic experimental design. Therefore, it is considered that it is okay to be published in the journal without any modification.

Author Response

Dear Reviewer
Enclosed you can find a copy of our revised Manuscript ID D foods-2462315, entitled “Microbial
and physico-chemical characterization of cold smoked sea bass (Dicentrarchus labrax), a new
product of fishery.
Journal: Foods
I add the answer to the referee.
The authors would like to thank the reviewers for their careful reading of the manuscript and the
resulting constructive comments and suggestions. Basically, we agree with all of the points raised
by the reviewers, and wherever possible the manuscript has been modified as recommended. All
reviewer comments are in black plain font, whereas our response is described in red plain font.
We have made the changes and corrections on the basis of the reviewer’s suggestions. We evaluated
the comments and prepared a point-by-point response to each one of them.
Reviewer 1
Dear Reviewer
Thanks for the consideration you have about our manuscript
Regards
Giuseppe Comi

Reviewer 2 Report

The authors analysed several samples of cold smoked sea bass (conserved at 4-8ºC) in five periods from 0 to 60 days. The microbial and physico-chemical data revealed to be a safe product and with good nutritional value.

As a global analysis the main issue found is that the text it is too dense too read. A lot of important information is presented but, in some cases, less words could be used to give that information. It is too “wordy” and sometimes it is difficult to keep the focus and keep reading.

The introduction will benefit if it is shortened. All points presented are important but there is some repetition especially related with the species present in CSF.

Minor comments

1.    Line 17 and line 772: “a new product that has not yet been commercialized”; “novel processed food product of Italian fisheries”. It is not clear to me in what stage it is the product. On line it also seems that a lot of recipes of “branzino affumicato a freddo” are available.

2.    Line 87 – “or as a delicacy commonly consumed as a 'ready-to-eat' (RTE) product without heat treatment [15].” This sentence it is out of context in this part of the text.

3.    Line 115: here you can use “lactic acid bacteria” or just “LAB” since the LAB was previously presented in line 95.

4.    Lines 115, 447, 642: Change “microflora” to microbiota

5.    Lines 394: Change “flora” to microbiota

6.    Lines 132-141, 461: Apply the italic format to the names of genera and species.

7.    Line 185: Change hours to h

8.    Line 229: Change MAE to MEA

9.    Line 214: Change 0.8-mm-thick to 0.8 mm thick

10. Lines 214-216, 713-715: When using several parentheses apply the format [xxx(yyy)] instead of (xxx(yyy)).

11. Line 229: Change “transplanted” to transferred

12. Line 283: “a minute”. Do you mean 1 min or a small period of time?

13. Line 290. Change “cm/sec” to cm/s

14. Lines 307-308: In the values of ÊŽ change the comma to a dot (ÊŽ=422.7)

15. Line 338: Change minutes to min

16. Line 181, 359, 466: Different number of days are presented? “stored for 30 days”, “for 20 days”, “for 2 days”

17. Line 400: “several studies [60].” Several but only one reference?

18. Lines 409-416: MPN can be converted to CFU/g and that it is the common analysis. Since in the work presented MPN analysis was not applied, how can the authors compare data?

19. Lines 529-530: “or indicates a lack of monitoring for this strain.” I did not understand what do you mean.

20. Lines 535-536 and 546-547 have the same data.

21. Lines 743-744: Change “by Kocatepe & Turan [83], Bhouri et al. [78, Fuentes 743 et al. [58] and Yildiz et al. [89].” to “by other authors [58,78,83,89].

.

Author Response

Dear reviewer

Enclosed you can find a copy of our revised Manuscript ID D foods-2462315, entitled “Microbial and physico-chemical characterization of cold smoked sea bass (Dicentrarchus labrax), a new product of fishery.

Journal: Foods

I add the answer to the referee.

The authors would like to thank the reviewers for their careful reading of the manuscript and the resulting constructive comments and suggestions. Basically, we agree with all of the points raised by the reviewers, and wherever possible the manuscript has been modified as recommended. All reviewer comments are in black plain font, whereas our response is described in red plain font.

We have made the changes and corrections on the basis of the reviewer’s suggestions. We evaluated the comments and prepared a point-by-point response to each one of them.

Reviewer 2

The authors analysed several samples of cold smoked sea bass (conserved at 4-8ºC) in five periods from 0 to 60 days. The microbial and physico-chemical data revealed to be a safe product and with good nutritional value.

As a global analysis the main issue found is that the text it is too dense too read. A lot of important information is presented but, in some cases, less words could be used to give that information. It is too “wordy” and sometimes it is difficult to keep the focus and keep reading.

The introduction will benefit if it is shortened. All points presented are important but there is some repetition especially related with the species present in CSF.

Answers – Thanks many parts of the paper have been shortened and improved and the repeats eliminated.

Lines 57-59 – eliminated - where fish is usually sold in boxes with ice flakes or at 4 °C packaged in air or under vacuum, both as whole gutted fish and as fillets,

Lines 61-64 – eliminated - 7 that [7]; aquaculture represents one of the major food production systems that has widely grown over the last decades due to the increasing demand for seafood caused by worldwide population growth and the increased awareness of its beneficial effects on human health [6]. Iin Europe, sea bass are

Lines 77-78- eliminated   In addition to drying, smoking is one of the oldest methods used for fish preservation [12].

Lines 87 -88 – eliminated - or as a delicacy commonly consumed as a 'ready-to-eat' (RTE) product without heat treatment [15].

Lines 97-99 – eliminated - Finally, in certain CSF products, mMould and yeast were also isolated, at a level of 4 log CFU/g and are recognized as presumptive spoilers [20]. However, they are

Lines 117-119 – eliminated -. In some CSF samples supplemented with herbs, the predominance of Enterobacteriaceae was observed, although LAB and, to a lesser extent, Brochothrix spp. were also present [17,28].

Line 121 – eliminated - such as 16S rRNA gene Illumina metabarcoding or 4-5-4 pyrosequencing …

Line 123 eliminated – foods

Lines 131 – 134 eliminated - Lactobacillus curvatus and Staphylococcus equorum are often the dominant species among LAB and cocci, respectively; other strains such as L. sakei, L. plantarum, Lactococcus lactis, C. divergens, C. maltaromaticum, C. piscicola and Leuconostoc spp. were isolated in smaller numbers [20].

Lines 142 – 144 – eliminated - B. thermosphacta, mould, and yeast were not considered real spoilers of CSF and were isolated in particular in the injection brined of Cold Smoked Salmon (CSS) but not in dry salted salmon [30].

Lines 157 – 158 eliminated - in order to contribute to the diversification of seafood products and satisfy consumers’ requests and preferences and sustainability needs,

Lines 159 – 160 – eliminated - ), a traditional Italian aquaculture fish.

Lines 449 – 462 – eliminated - In fact, certain psychrotrophic pathogenic microorganisms can contaminate CSF during the smoking process. Contamination originates from the environment, handling and processing [60]. Listeria monocytogenes is a psychrotrophic, nonspore-forming bacterium that can cause listeriosis [12,23,24]. Cold smoked fish can be a source of foodborne listeriosis [26]. The prevalence of L. monocytogenes in retail CSS is highly variable and varies from 0 to 100% [16]. Variations in sampling, particularly in the time of year, handling, and analytical methods, are also likely to contribute significantly to the variation [12]. L. monocytogenes is able to survive smoking and the salting steps performed during the production process of CSS [65] and to grow well both aerobically and anaerobically (vacuum-packed) at refrigerated temperatures [4]. For this reason, it was suggested to inhibit its growth using bioprotective cultures. Starter cultures could not completely inhibit L. monocytogenes, but they could slow its growth. As a result, the L. monocytogenes count exceeds 2 log CFU/g, exposing consumers to the risk of listeriosis [4].

Lines 544 – 548 – eliminated - Our data showed that TVB-N increased from initial values of 28.2 mg N/100 g to 32.2 at 30 days and to 34.3 mg N/100 g at the end of the shelf life. Significant differences were not observed until day 15; then, the increase was significantly different, demonstrating that the abuse temperature is fundamental for microbial growth, tissue enzyme activation, and the increase in TVB-N.

Lines 620 – 621 – eliminated - Psychrotrophic Enterobacteriaceae are often isolated in CSF, and in some cases, they can dominate until the end of the shelf life of the product [20].

Lines 649 – 652 – eliminated - In addition, some authors have argued that several bacteria, including B. thermospacta, Aeromonas spp., and, in some cases, mould and yeast [20], have been isolated from spoiled CSS, indicating a relatively complex and variable microbial population in this type of product.

Minor comments

  1. Line 17 and line 772: “a new product that has not yet been commercialized”; “novel processed food product of Italian fisheries”. It is not clear to me in what stage it is the product. On line it also seems that a lot of recipes of “branzino affumicato a freddo” are available.

Answer: Thanks – Yes, you are right  - I change with –

Lines 17 - a novel fish product of  Italy

Lines 772 - is a processed food product of Italian fishery.

  1. Line 87 – “or as a delicacy commonly consumed as a 'ready-to-eat' (RTE) product without heat treatment [15].” This sentence it is out of context in this part of the text.

Answer - I eliminate it – Lines 86 - or as a delicacy commonly consumed as a 'ready-to-eat' (RTE) product without heat treatment [15].

  1. Line 115: here you can use “lactic acid bacteria” or just “LAB” since the LAB was previously presented in line 95.

Answer – Thanks - I correct it -  Line 114 - LAB

  1. Lines 115, 447, 642: Change “microflora” to microbiota

Answer – Thanks - Lines 114 – 445 – 637 – I change microflora with microbiota 

  1. Lines 394: Change “flora” to microbiota

Answer – Thanks-  Line 392 - I change flora with microbiota

  1. Lines 132-141, 461: Apply the italic format to the names of genera and species.

Answer – Thanks - I eliminate the sentence because it is a repetition – Lines 130 – 133 Lactobacillus curvatus and Staphylococcus equorum are often the dominant species among LAB and cocci, respectively; other strains such as L. sakei, L. plantarum, Lactococcus lactis, C. divergens, C. maltaromaticum, C. piscicola and Leuconostoc spp. were isolated in smaller numbers [20]. 

Lines 459-461 – the sentence was eliminated - Starter cultures could not completely inhibit L. monocytogenes, but they could slow its growth. As a result, the L. monocytogenes count exceeds 2 log CFU/g, exposing consumers to the risk of listeriosis [4].

  1. Line 185: Change hours to h

Answer – Thanks - I correct – Line 183 – 72 h.

  1. Line 229: Change MAE to MEA

Answer – Thanks - I correct it – Line 227 – MEA

  1. Line 214: Change 0.8-mm-thick to 0.8 mm thick

Answer – Thanks – Line 212 - I made it, 0.8 mm thick

  1. Lines 214-216, 713-715: When using several parentheses apply the format [xxx(yyy)] instead of (xxx(yyy)).

Answer – Thanks - I correct it – Lines 212 – 214 [8% (wt/vol) acrylamide bisacrylamide (37.5:1)), with a denaturing gradient from 30% to 50% (100% corresponded to 7 M urea and 40% (wt/vol) formamide]

Lines 708 – 710 - [phosphorus (P), potassium (K), sodium (Na), calcium (Ca) and magnesium (Mg)] and microelements [iron (Fe), zinc (Zn) and copper (Cu)],

  1. Line 229: Change “transplanted” to transferred

Answer – Thanks - I change it – Line 227 - transferred

  1. Line 283: “a minute”. Do you mean 1 min or a small period of time?

Answer – Thanks – I mean one minute – Line 281 – one  minute

  1. Line 290. Change “cm/sec” to cm/s

Answer – Thanks – I correct it – Line 288 - 43 cm/s.

  1. Lines 307-308: In the values of ÊŽ change the comma to a dot (ÊŽ=422.7)

Answer – Thanks – Lines 305 – 306 – I correct it - at ÊŽ=589.0 nm for Na, ÊŽ=422.7 nm for Ca, ÊŽ=766.5 nm for K, ÊŽ=285.2 nm for Mg, ÊŽ=324.8 nm for Cu, ÊŽ=213.9 nm for Zn, and ÊŽ=248.3 nm for Fe

  1. Line 338: Change minutes to min

Answer - Thanks – I correct it – Lines 336 – 13 min

  1. Line 181, 359, 466: Different number of days are presented? “stored for 30 days”, “for 20 days”, “for 2 days”

Answer – Thanks – I am wrong, I correct - Lines 179 – 357 – 464  

at 4 ± 2 °C for 20 days and then at 8 °C till 60 days.

  1. Line 400: “several studies [60].” Several but only one reference?

Answer – Thanks – I correct it – Lines 397 – 398 - Enterobacteriaceae can grow and reach high loads, as demonstrated by Fuentes et al., (60).

  • . Lines 409-416: MPN can be converted to CFU/g and that it is the common analysis. Since in the work presented MPN analysis was not applied, how can the authors compare data?

Answer – Thanks – Usually the traditional laboratories of microbial analysis consider the value of MPN as CFU/g. However, I correct the sentence with Lines 410 – 413 - Considering the different Enterobacteriaceae method of analysis, it was impossible to use the Sernapesca [55] limits to value the acceptability of our investigated samples.  

  1. Lines 529-530: “or indicates a lack of monitoring for this strain.” I did not understand what do you mean.

Answer – Thanks – I try to explain and I change the sentence – Lines 527 – 529 - In the current study, all the tested samples had a WPS value greater than 5.0%, consequently they did not support the C. botulinum growth.

  1. Lines 535-536 and 546-547 have the same data.

Answer – Thanks – I eliminate the sentence – Lines 544 – 548 - Our data showed that TVB-N increased from initial values of 28.2 mg N/100 g to 32.2 at 30 days and to 34.3 mg N/100 g at the end of the shelf life. Significant differences were not observed until day 15; then, the increase was significantly different, demonstrating that the abuse temperature is fundamental for microbial growth, tissue enzyme activation, and the increase in TVB-N

  1. Lines 743-744: Change “by Kocatepe & Turan [83], Bhouri et al. [78, Fuentes 743 et al. [58] and Yildiz et al. [89].” to “by other authors [58,78,83,89].

Answer – Thanks – I correct it – Lines 739 - but higher than the data obtained by other authors [58,78,83,89].
